# Rhodium (II)-Catalyzed Synthesis of Tetracyclic 3,4-Fused Indoles and Dihydroindoles

**Hongwei Qiao** [1,†], **Jiakun Bai** [2,†], **Mengyao Zhu** [2], **Juanhong Gao** [2], **Anna** [2], **Sichun Zhang** [1,*] **and Chao Li** [2,*] 

1   Department of Chemistry, Tsinghua University, Beijing 100084, China; qhw15@mails.tsinghua.edu.cn
2   State Key Laboratory of Chemical Resource Engineering, Beijing University of Chemical Technology, Beijing 100029, China; jiakun_bai@163.com (J.B.); zhumengyao815@163.com (M.Z.); gjuanhong@163.com (J.G.); anna@mail.buct.edu.cn (A.)
*   Correspondence: sczhang@mail.tsinghua.edu.cn (S.Z.); lichao@mail.buct.edu.cn (C.L.)
†   H.Q. and J.B. contributed equally to this work.

**Abstract:** An efficient synthetic method of tetracyclic 3,4-fused indoles and dihydroindoles via rhodium-catalyzed (3+2) cycloaddition of *N*-tosyl-4-(2-phenoxyphenyl)-1,2,3-triazole was described. The aromatized xanthene derivatives can be achieved in a one-pot synthesis starting from 1-ethynyl-2-phenoxybenzene. The xanthene-based fused heterocycles were considered as the valuable fluorophore.

**Keywords:** rhodium catalysis; xanthene; cycloaddition; triazole; fluorophore

## 1. Introduction

Xanthene-based fluorescent dyes largely containing fluorescein and rhodamine have attracted continuous attention from researchers because of their good photophysical properties such as high absorption coefficient, high photostability, and high fluorescence quantum yield [1–8]. However, absorption and emission wavelengths of many xanthene derivatives are in the ultraviolet-visible light range below 600 nm, which makes them unsuitable for bioimaging in living systems [9]. An important modification to the dyes is the introduction of a fused aryl ring into the xanthene skeleton. This modification brings a remarkable bathochromic shift in excitation and emission wavelengths [10]. The classic examples are Rho 101, naphthoxanthene, and SNARF-1, which exhibit much longer wavelengths than those of rhodamine and fluorescein under basic conditions (Figure 1) [11–13].

Rhodium-stabilized donor/acceptor carbenes as reactive intermediates have been widely applied in modern organic synthesis [14–26]. Among them, *N*-sulfonyl-1,2,3-triazole as an alternative source of carbene precursor has been used to achieve the transannulation reaction for the direct synthesis of heterocycles [27–41]. Murakami's group described a rhodium-catalyzed (3+2) annulation reaction of tricyclic 3,4-fused dihydroindoles via the corresponding 1,2,3-triazoles [42–45]. Recently, Davies et al. described a series of rhodium(II)-catalyzed intramolecular annulations of indolyl- and pyrrolyl-tethered *N*-sulfonyl-1,2,3-triazoles, including tetrahydropyrrolopyridine, tetrahydrocarboline, tetrahydro pyrrolo-[2,3-*d*]azepine, and azepino [4,5-*b*]indoles [46,47]. Based on these findings, we wondered if an analogous α-imino carbene can be used to construct a tetracyclic aryl-fused structure from simple materials in the presence of rhodium(II) catalysts. Thus, we here described a rhodium-catalyzed intramolecular (3+2) annulation to synthesize tetracyclic compounds. In addition, the reaction constituted a simple synthesis of aromatizing pyrrole-fused xanthene starting from 1-ethynyl-2-phenoxybenzene in one pot (Figure 1).

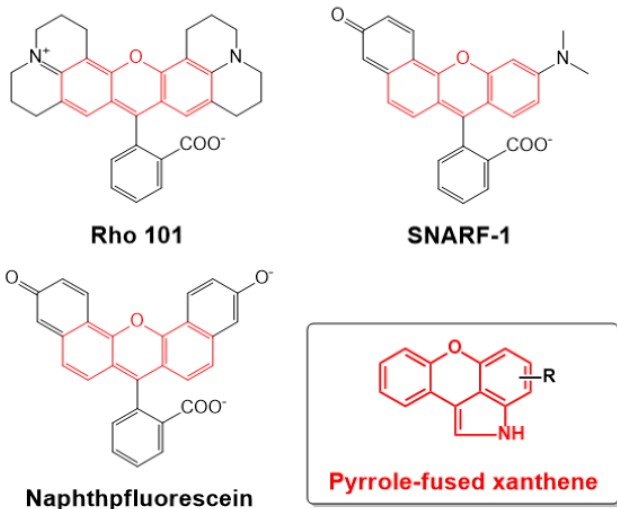

**Figure 1.** The fluorophores based on xanthene moiety.

## 2. Results and Discussion

Firstly, the substituted phenylboronic acids were coupled with 2-iodophenol in the presence of Cu(OAc)$_2$ to provide 1-iodo-2-phenoxybenzene **1**. Iodine was converted to a trimethylsilylacetylene (TMSA) functional group using PdCl$_2$(PPh$_3$)$_2$ and CuI as the catalysts, and then trimethylsilyl was removed in CH$_2$Cl$_2$ solution containing K$_2$CO$_3$ to give 1-ethynyl-2-phenoxybenzene **3** [48]. Subsequently, the key intermediate *N*-tosyl-4-(2-phenoxyphenyl)-1,2,3-triazole **4** was obtained via (3+2) cycloaddition of **3** with tosyl azide in the presence of CuTc. When the triazole **4** reacted with rhodium(II) octanoate dimer in toluene at 80 °C for 4 h, the desired tetracyclic compound **5** was obtained after chromatography purification. The results delineated the scope of the (3+2) annulation reaction as shown in Scheme 1. Substrates possessing the electron-withdrawing groups smoothly reacted, and the corresponding products **5c–5f** were isolated in yields ranging from 84% to 92%. The unsubstituted and electron-donating group substituted substrates also successfully involved in the transformation with yields of 72% and 70%, respectively (**5a** and **5b**). The stereochemistry of the tetracyclic products was analyzed by $^1$H NMR data of **5a–5f**. H atoms on the methenyl group showed relatively large coupling constants (*J* = 14.0–15.8 Hz). Narasaka K. et al. reported a series of *cis*- and *trans*-fused (4,5,6,7-*η*)-3a,7a-dihydro-3*H*-indoles, in which the *J* of the *cis*-fused isomer was much larger than the *trans*- one [49]. Similarly, the coupling constants of *cis*-fused indole derivatives in Murakami's report were also up to 14 Hz [42]. Thus, the compounds **5a–5f** were considered as *cis*-fused isomers.

**Scheme 1.** *Cont.*

**5a** 72%   **5b** 70%   **5c** 85%

**5d** 92%   **5e** 84%   **5f** 86%

**Scheme 1.** Tetracycles synthesis by Rh(II) catalyst [a]. [a] Reaction conditions: (i) 2-iodophenol (1.0 equiv), substituted arylboronic acids (1.5 equiv), $Et_3N$ (5.0 equiv), anhydrous $Cu(OAc)_2$ (1.2 equiv), $CH_2Cl_2$, r.t., 6 h. (ii) **1** (1.0 equiv), TMSA (1.1 equiv), $PdCl_2(PPh_3)_2$ (0.02 equiv), CuI (0.04 equiv), $Et_3N$, r.t., 4 h. (iii) **2** (1.0 equiv), $K_2CO_3$ (0.5 equiv), $CH_2Cl_2$:$CH_3OH$ = 1:1, r.t., 2 h. (iv) **3** (1.0 equiv), $TsN_3$ (1.0 equiv), TcCu (0.05 equiv), toluene, r.t., $N_2(g)$, 12 h. (v) **4** (1.0 equiv), $Rh_2(oct)_4$ (0.02 equiv), toluene, 80 °C, $N_2(g)$, 4 h.

We next speculated the possible mechanism for the production of the tetracyclic 3,4-fused dihydroindole **5** (Scheme 2). An $\alpha$-diazo imino **A** was formed by reversible tautomerization from the corresponding triazole **4**. The intermediate $\alpha$-imino rhodium carbene **B** was obtained with the release of $N_2$ (*g*) when **A** rapidly reacted with rhodium (II). Subsequently, the intramolecular electrophilic reaction of **B** occurred to form the zwitterionic intermediate **C**. The anionic rhodium released bonding electrons, completing the second cyclization [42]. The intermediate **A** acts as a 1,3-dipole equivalent due to the nucleophilic character of the imino nitrogen. According to Davies report [47], alternatively, $\alpha$-diazoimine **A** undergoes thermal decomposition to generate free carbene **D**, which could cyclopropanate an arene to give norcaradiene **E**. The intermediate **E** does not have the correct geometry to undergo a (3,5)-sigmatropic rearrangement to give **5**.

**Scheme 2.** Proposed mechanistic pathways.

It is noted that it was a bit hard to isolate compound **5** after the reaction because we found these tetracyclic structures were easily oxidized in air, giving the corresponding aromatized pyrrole-fused xanthene derivative **6**. According to Murakami's report, a special oxidant (such as $MnO_2$) was needed to complete further oxidative aromatization [42]. However, in our case, thorough aromatization can be achieved by stirring in air for some hours. Thus, we subsequently tried to directly synthesize aromatized xanthene derivatives.

The construction of these pyrrole-fused xanthene compounds was successfully integrated into a one-pot synthesis directly from 1-ethynyl-2-phenoxybenzene **3** (Scheme 3). For example, **3g** (1.0 equiv),

tosyl azide (1.0 equiv), CuTc (0.05 equiv), $Rh_2(oct)_4$ (0.02 equiv), and toluene were mixed together in a flask. The above mixture was stirred at 25 °C for 12 h, during which **3g** was converted to the corresponding triazole **4g**. The mixture was then stirred at 80 °C for additional 4 h. After being cooled to room temperature, the mixture was further stirred for 4 h in air. Finally, preparative thin-layer chromatography was used to afford **6g** in 73% yield based on **3g**. The all-in-one-pot procedure showed that the catalysts and reagents requisite in each step barely interfered with each other.

**Scheme 3.** One-pot synthesis starting from 1-ethynyl-2-phenoxybenzene.

The present reaction was also used to synthesize xanthone **7** from triazole diaryl ether **4**, as shown in Scheme 4. When *N,N*-diethyl substituted **4k** in toluene and was catalyzed by $Rh_2(oct)_4$ at 80 °C, no pyrrole-fused xanthene skeleton was obtained. Instead, the corresponding xanthone **7k** was obtained in 75% isolated yield. After the first cycloaddition to form intermediate **C**, the strong electron-donating substituent $N(C_2H_5)_2$ weakened the electropositivity of the allyl position, which was unfavorable for the second cyclization with the imino nitrogen. The tosyl amide was further oxidized to a carbonyl group in air, affording a stable xanthone derivative. Similarly, the substrate **4l'** with an intense electron-donor group diethyl amine also underwent a single cycloaddition reaction to give xanthone **7l'** in 80% yield. Shen et al. reported a similar process for the synthesis of *N*-methyl acridone derivatives [50]. The triazole intermediate was converted to acridone by rhodium catalysis via a single cycloaddition.

**Scheme 4.** Synthesis of 9H-xanthen-9-one [a].

In addition, a special case of the rhodium catalysis process was obtained for the *O*-methoxy substrate. The (3+2) annulation reaction of *N*-tosyl-4-(2-phenoxyphenyl)-1,2,3-triazole **4m** took place by $Rh_2(oct)_4$ in toluene under the same conditions. However, oxidative aromatization directly occurred during the process of annulation, and also the *O*-methoxy group was simultaneously removed, affording the aromatized pyrrole-fused xanthene **8** with 78% yield (Scheme 5). Due to the rotatable C-O bond in diaryl ether, the carbenoid carbon of **B** is electrophilic to react with methoxyl site. When intramolecular attack of the phenyl ring in the methoxyl site occurs to furnish the zwitterionic intermediate, methoxyl as a good leaving group could be removed. In the tested substrates, only *O*-methoxy-substituted triazole could be aromatized, retaining a tosyl group in the oxidation process.

**Scheme 5.** Special oxidative aromatization reaction of 4-(2-phenoxyphenyl)-1,2,3-triazole.

Encouraged by the straightforward synthetic pathway described above, we tried to investigate the fluorescent properties of the pyrrole-fused xanthene skeleton. The excitation ($\lambda_{ex}$) and emission ($\lambda_{em}$) wavelengths of the typical structures were tested as shown in Table 1. The unaromatized **5a** showed spectral characteristics that were comparable to rhodamine with $\lambda_{em} = 443$ nm and $\Phi = 0.31$. The entirely aromatized structures via oxidation, such as **6g–6j**, exhibited longer emission wavelengths at 471, 480, and 552 nm, respectively. When the substituent was an electron-withdrawing $NO_2$ group, the fluorescent properties including emission wavelength and quantum yield remarkably increased. Two xanthone-based products, **7k** and **7l′**, were also analyzed, giving $\lambda_{em} = 426$ nm and $\lambda_{em} = 457$ nm with acceptable quantum yields, respectively. These results indicated that the pyrrole-fused xanthene or imino-modified derivatives in pyrrole were used as a potential fluorophore to develop new applications.

**Table 1.** Fluorescent properties of the typical structures [a].

| Comp. | Solvent | $\lambda_{ex}$ (nm) | $\lambda_{em}$ (nm) | Quantum Yield ($\Phi$) |
|---|---|---|---|---|
| **5a** | $CH_2Cl_2$ | 388 | 443 | 0.31 |
| **6g** | $CH_3CN$ | 370 | 471 | 0.45 |
| **6h** | $CH_2Cl_2$ | 327 | 480 | 0.38 |
| **6j** | DMSO | 436 | 552 | 0.52 |
| **7k** | DMSO | 355 | 426 | 0.41 |
| **7l′** | DMSO | 355 | 457 | 0.35 |

[a] See SI Figure S1 for details of emission ($\lambda_{em}$) wavelength and quantum yield ($\Phi$) for the typical compounds.

## 3. Materials and Methods

### 3.1. Materials

Unless specifically mentioned, all chemicals were purchased from Beijing Ouhe Technology Co. Ltd., Beijing, China, or J&K Scientific Ltd., Beijing, China, and used without further purification.

### 3.2. Typical Procedure for the Synthesis of Triazole **4**

A mixture of **3a** (1.32 g, 6.80 mmol) and copper(I) thiophene-2-carboxylate (64.79 mg, 0.34 mmol) was dissolved in dry toluene (30 mL). $TsN_3$ (1.38 mL, 6.80 mmol) was added via syringe, and the solution was stirred at room temperature for 12 h under a nitrogen atmosphere. The crude product was

further recrystallized from hexane/$CH_2Cl_2$ (15:1) to yield the final pure **4a** as a white solid (77% yield). Compounds **4b–f** were synthesized using a similar route according to **4a**.

### 3.3. Tetracycle **5** Synthesis by Rh(II) Catalyst

Compound **4a** (0.65 g, 1.66 mmol) and $Rh_2(oct)_4$ (25.86 mg, 0.03 mmol) were dissolved in dry toluene (10 mL) in a Schlenk tube. The solution was heated for 4 h at 80 °C. Then, the mixture was evaporated under vacuum to give the crude product, which was purified by silica gel column chromatography hexane/ethyl acetate (20:1) to give **5a** (72% yield). Compounds **5b–f** were synthesized using a similar route according to **5a**.

### 3.4. One-Pot Synthesis of **6** Starting from 1-ethynyl-2-phenoxybenzene

Compound **3g** (0.8 g, 3.84 mmol), $TsN_3$ (0.76 g, 3.84 mmol), CuTc (36.62 mg, 0.19 mmol), and $Rh_2(oct)_4$ (59.82 mg, 76.83 μmol) were dissolved in dry toluene (10 mL) in a Schlenk tube. The solution was stirred at room temperature for 12 h and then heated to 80 °C for 4 h. After being cooled to room temperature, the solution was further stirred at room temperature for 4 h in air. Then, the mixture was evaporated under vacuum to give the crude product, which was purified by silica gel column chromatography hexane/ethyl acetate (20:1) to give **6g** (60% yield). Compounds **6h–j** were synthesized using a similar route according to **6g**.

### 3.5. Synthesis of 9H-xanthen-9-one and Special Oxidative Aromatization Reaction of 4-(2-phenoxyphenyl)-1,2,3-triazole

Compounds **7k**, **7l′,** and **8** were synthesized using a similar route according to **5a**.

### 3.6. Calculation of the Fluorescence Quantum Yield

The quantum yield of the fluorophore was calculated according to Equation (1):

$$\varphi_U = \varphi_S \left(\frac{F_U}{F_S}\right)\left(\frac{A_S}{A_U}\right)\left(\frac{\eta_U^2}{\eta_S^2}\right) \tag{1}$$

where $\varphi_s$ is the quantum yield of the standard, $F$ is the area under the emission spectra, $A$ is the absorbance at the excitation wavelength, and $\eta$ is the refractive index of the solvent used. $U$ subscript denotes unknown, and $S$ means standard. Rhodamine B was chosen as the standard.

## 4. Conclusions

In conclusion, we have described the intramolecular (3+2) annulation of $\alpha$-imino rhodium carbene complexes to construct tetracyclic 3,4-fused indoles and dihydroindoles. Of note is that the reaction is illustrated by its successful integration into a one-pot synthesis directly from 1-ethynyl-2-phenoxybenzene, giving a series of xanthene derivatives. Xanthenes are important in fluorescent dye and medicinal chemistry, and we think that the current approach is an appealing choice for the construction of molecular libraries for diversity-oriented synthesis.

**Supplementary Materials:** The following are available online at http://www.mdpi.com/2073-4344/10/8/920/s1. NMR and HRMS data of compounds **4–8**; [1]H NMR, [13]C NMR, and HRMS spectra of compounds **4–8**; fluorescence spectra of **5a**, **6g**, **6h**, **6j**, **7k,** and **7l′**.

**Author Contributions:** Performed the experiments, H.Q. and J.B.; wrote the manuscript, S.Z. and C.L.; analyzed the characterization of compounds, M.Z., J.G., and A. All authors have read and agreed to the published version of the manuscript.

**Funding:** This research was funded by the National Natural Science Foundation of China (21974078, 21672021, and 21572018).

**Conflicts of Interest:** The authors declare no conflict of interest.

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
