# Peer review of "Rhodium (II)-Catalyzed Synthesis of Tetracyclic 3,4-Fused Indoles and Dihydroindoles"

_catalysts, doi:10.3390/catal10080920_

Round 1

Reviewer 1 Report

The presented paper is devoted to the study on the rhodium-catalyzed [3+2] intramolecular cycloaddition in N-tosyl-4-(2-phenoxyphenyl)-1,2,3-triazoles that gave various types of products depending on the nature of substituents in the aromatic ring. The aim of the work was the synthesis of pyrrole-fused xanthene derivatives that could be used for the development of new fluorescent dyes.

I suppose that the paper is not suitable for publication in the present form due to several reasons:

  1. The article is carelessly written; the material is poorly structured and dumped in a heap. In fact, it turned out that the direction of the reaction significantly depends on the nature of the substituent in the aromatic ring. The authors sought to synthesize structures of type 6, but the structures were obtained only in some cases both in the presence of EWG and EDG groups in m- and p-positions in the aromatic ring. Unfortunately, there is no clear explanation for this effect.
  2. It is not clear how the structures 6 are formed at all, because the classical [3+2] addition should give structures 5, for which, by the way, stereochemistry is not shown and proven by the authors. Compounds 5 are not xanthenes.
  3. Why substrates 4k and 4l’ provide xanthen-9-ones? For this product, the C-C bond oxidative cleavage should occur. There is no explanation as well.
  4. It is also not clear why different methods of synthesis (stepwise or one-pot) produce different products - 5 or 6?
  5. There is only one table in the manuscript – Table 4, another “tables” are schemes or figures.
  6. Line 4 – Who is Anna2?
  7. Lines 48-52 – Where is no reference on the synthesis of compound 3.
  8. The experimental part is poorly presented. The figures with spectra in SI should be enlarged. NMR spectra were recorded on the spectrometer, not spectroscopy. There is no information on H-F and C-F coupling constants for compounds 4e, 4f, 5e, 5f. There are no fluorescence spectra in SI as it was mentioned in Table 4.
  9. The English text must be carefully proofread because there are many typos and grammatical errors. For example, “pyrrole-refused”, “4k was catalyzed”, “compounds was successful integrated” etc.

Reviewer 2 Report

The manuscript describes an efficient synthetic method of pyrrole-fused xanthene derivatives via rhodium-catalyzed [3+2] cycloaddition of N-tosyl-4-(2-phenoxyphenyl)-1,2,3-triazole derived from 1-ethynyl-2-phenoxybenzene. In this work, the authors developed the intramolecular [3+2] annulation of α-imino rhodium carbene complexes to construct pyrrole-fused xanthene skeleton whose derivatives are considered as a valuable fluorophore. In addition, they succeeded in a one-pot synthesis of pyrrole-fused xanthenes starting from 1-ethynyl-2-phenoxybenzene. Furthermore, they found the present [3+2] annulation reaction was also used to synthesize xanthones from the corresponding triazole diaryl ethers by introducing N,N-diethyl group on benzene rings. This method could potentially serve as diversity-oriented synthetic methods for xanthene derivatives.

Although it seems there is room for improvement in yield of one-pot synthesis, overall the authors present a nice study that describes the synthesis and applications. This manuscript is potentially suitable for publication in Catalysis.

Author Response

Thank you for the useful comments. We will try to improve yield of one-pot synthesis in the future work.

Reviewer 3 Report

Authors report the synthesis of xanthene derivatives via rhodium catalyzed cycloaddition. In my opinion the Message the authors want to convey with their paper is clear, however i believe their claim that the synthesis of the xanthene compounds as fluorescent probes should be better specified And clarified:

1) Which applications are they thinking of? Do they mention sensing purposes?

2)Which analytes could be sensed and by Which mechanism the recognition event should occurr?

3) preliminary fluorescent experiments would help the reader To better understand the Final aim of the work.

Round 2

Reviewer 1 Report

The authors did a good job on the article correction. However, many moments remain incomprehensible.

The title of the work and the conclusions still do not correspond to the content. Only four compounds among the final 13 products are xanthenes.

The stereochemistry of compounds 5 must be proven. To do this, the authors just need to study NMR spectroscopy.

For compounds 4e, 4f, 5e, 5f the H-F and C-F constants should be given. For this, an NMR spectrometer on 400 (1H) and 100 (13C) MHz is sufficient. The point is not in the spectrometer resolution but in the qualification of the authors, who believe that it is enough to simply list the lines in the spectrum. Moreover, for compounds 4, no mass spectrometry data are given at all.

The fluorescence spectra must be given for all compounds presented in Table 1.

What does it mean to carry out the reaction in the fresh air? And if you use stale air, what happens?

The Schlenk tube must be capitalized. This name is in honor of the scientist who invented the technology of working in an inert atmosphere (https://en.wikipedia.org/wiki/Wilhelm_Schlenk).

Round 3

Reviewer 1 Report

I recommend removing word “skeletons” from the title. The next title could be appropriate: “Rhodium(II)-catalyzed synthesis of tetracyclic 3,4-fused indoles and dihydroindoles”

The constants J for fluorinated compounds should be written as JHH and JHF, as well as JCF

Lines 77-82: the text must be rewritten - it repeats the text in the Ref. [13a]. An explanation of the cis-configuration should be given in the text with appropriate references. By the way, XRD is carried out in the same reference [13a] and cis- constants are also 14 Hz.
